# Proteins of the Nucleolus of *Dictyostelium discoideum*: Nucleolar Compartmentalization, Targeting Sequences, Protein Translocations and Binding Partners

**DOI:** 10.3390/cells8020167

**Published:** 2019-02-17

**Authors:** Danton H. O’Day

**Affiliations:** 1Department of Biology, University of Toronto Mississauga, Mississauga, ON L5L 1C6, Canada; danton.oday@utoronto.ca; Tel.: +905-808-6566; 2Department of Cell and Systems Biology, University of Toronto, Toronto, ON M5S 3G5, Canada

**Keywords:** nucleolus, *Dictyostelium*, nucleolar localization sequences, protein translocation, closed mitosis, disease

## Abstract

The nucleoli of *Dictyostelium discoideum* have a comparatively unique, non-canonical, localization adjacent to the inner nuclear membrane. The verified nucleolar proteins of this eukaryotic microbe are detailed while other potential proteins are introduced. Heat shock protein 32 (Hsp32), eukaryotic translation initiation factor 6 (eIF6), and tumour necrosis factor receptor-associated protein 1 (TRAP1) are essential for cell survival. NumA1, a breast cancer type 1 susceptibility protein-C Terminus domain-containing protein linked to cell cycle, functions in the regulation of nuclear number. The cell cycle checkpoint kinase 2 homologue forkhead-associated kinase A (FhkA) and BRG1-associated factor 60a homologue Snf12 are also discussed. While nucleoli appear homogeneous ultrastructurally, evidence for nucleolar subcompartments exists. Nucleolar localization sequences (NoLS) have been defined that target proteins to either the general nucleolar area or to a specific intranucleolar domain. Protein translocations during mitosis are protein-specific and support the multiple functions of the *Dictyostelium* nucleolus. To enrich the picture, binding partners of NumA1, the most well-characterized nucleolar protein, are examined: nucleolar Ca^2+^-binding protein 4a (CBP4a), nuclear puromycin-sensitive aminopeptidase A (PsaA) and Snf12. The role of *Dictyostelium* as a model for understanding the contribution of nucleolar proteins to various diseases and cellular stress is discussed throughout the review.

## 1. Introduction

The nucleolus is a multifunctional subnuclear compartment that has been studied for more than 200 years [1,2]. More than 4500 proteins comprise the human nucleolar proteome of which approximately 30% function in ribosome biogenesis. Historically, studies have primarily focused on the transcription of rDNA genes and rRNA processing leading to the assembly of ribosomal subunits that are exported to the cytoplasm. These functions underlie the basic structure and organization of eukaryotic bipartite or tripartite nucleoli. Tripartite nucleoli exhibit a fibrillar center (FC; transcriptionally inactive rDNA), a dense fibrillar component (DFC; transcriptionally active rDNA plus rRNA processing), and a granular component (GC; ribosomal subunit assembly). Bipartite nucleoli have overlapping FC and DFC plus a GC region. During mitosis the nucleolus disassembles during prophase and reassembles during telophase. Despite the early interest in ribosome biogenesis, about 70% of the nucleolar proteome functions in other events including cell signaling, centrosome function, chaperone activity, DNA replication and repair, molecular sequestration, regulation of cell cycle events, stress response regulation and viral replication [3,4]. This diversity of functions underlies the central role of the nucleolus in a diversity of human diseases [1,2].

*Dictyostelium* excels as a model biomedical research organism for a multitude of reasons. It is inexpensive and easy to culture with a one-day asexual, developmental life-cycle. Possessing a haploid genome facilitates the generation of mutants by a diversity of molecular techniques. These and other strains and vectors plus multiple other resources are available from the Dicty Stock Center at dictybase.org. The separation of growth and development with comparatively simple differentiation facilitate the study of many fundamental cellular processes including cell growth, cell death, cytokinesis, cell movement, chemotaxis, mitosis, phagocytosis, as well as morphogenesis and differentiation [5]. In the last decade or so, *Dictyostelium* has gained prominence for the study of cell stress as well as human diseases including Batten’s disease, host-pathogen interactions, and Huntington’s disease [6].

## 2. The *Dictyostelium* Nucleolus

As in other eukaryotes, the multiple nucleoli are the largest intranuclear bodies in *Dictyostelium discoideum*. Early research revealed that the structural features of this social amoebozoan’s nucleoli differ from the classic nucleolar organization. Rather than localizing within the nucleoplasm, they exist as two to four dense patches that are tightly adhered to the inner nuclear envelope (Figure 1) [7,8,9,10,11]. What’s more, they are neither bipartite nor tripartite. Instead, ultrastructurally they present as a more-or-less homogenous structure consisting of continuous fibrous matrices within which different-sized ribosome-like granules (10–50 kDa) are distributed [12]. Fitting with the absence of defined FC regions, the rDNA instead forms a beaded ring-like structure (15–20 beads/ring) around the periphery of each nucleolus [13]. This rDNA is predominantly extrachromosomal with some being telomeric [5,14]. Other related amoebozoan species, including *D. mucoroides*, *D. minutum*, and *Polysphondylium pallidum*, appear to share the same nucleolar structure (e.g., [15,16]). In keeping with its role in rRNA synthesis in all species, treatment with actinomycin-D, an inhibitor of RNA polymerase I (i.e., rDNA transcription), leads to nucleolar breakdown in *Dictyostelium* [7,17].

This breakdown occurs in one of two ways: the progressive disappearance of protein localization (e.g., NumA1) or the formation of nucleolar buds—containing specific proteins (e.g., Cbp4a, Snf12, and FhkA)—that are released intact into the cytoplasm [18].

## 3. Nucleolar Subcompartments in *Dictyostelium*

Recent immunolocalization studies have shown that, despite earlier ultrastructural studies, the *Dictyostelium* nucleolus is not homogeneous (Figure 2) [18]. The nucleolar proteins of *Dictyostelium* organize as one of six observed patterns: localization to both the nucleolus and nucleoplasm (e.g., NumA1, eIF6, and Bud31), to the whole nucleolus (e.g., TRAP1) or to one of four subcompartments (NoSC1-4). CBP4a localizes to a patch close to the nuclear envelope designated as nucleolar subcompartment 1 (NoSC1). Snf12 localizes in NoSC2, a small speckle within NoSC1.

The site of rDNA localization at the nucleolar periphery (NoSC3) coincides with general distribution of two nucleolar proteins, Hsp32 and FhkA. The localization of Src1, a helix-extension-helix family homolog, may be a nucleolar protein so until verified as one, subcompartment NoSC4 remains in question. This compartmentalization suggests there is more to the structure and function of the nucleus than has historically been recognized. The question remains as to whether each of these designated regions contain functionally related proteins.

The discovery of nucleolar subcompartments should permit researchers to define function-specific domains within the nucleolus to answer that question [18]. There are multiple sources of evidence that support the presence of nucleolar subcompartments in *Dictyostelium*. First, specific nucleolar proteins routinely localize to specific regions of the nucleolus (Figure 2). Treatment with actinomycin D results in two defined but distinct patterns of nucleolar protein departure: the loss of individual proteins that disperse through the nucleoplasm and/or cytoplasm (e.g., NumA1) and the formation of nucleolar buds enriched in specific proteins (e.g., CBP4a, Snf12, and FhkA). Finally, the NLS/NoLS from Snf12 (KRKR) specifically localizes GFP to NoSC2 and thus represents the first nucleolar subcompartment localization signal (NoSCLS) identified in *Dictyostelium*.

This is not to say the *Dictyostelium* nucleolus is a static region dominated by rigid subcompartments. Each of the nucleolar proteins shows different degrees of variability in their localization which fits with the work of others showing the size and shape of nucleoli change with varying conditions ]10]. However, the stage has been set to examine the significance, constancy and regulation of nucleolar subcompartmentalization. If we examine the general function of the nucleolar proteins that have been identified to date, the primary overlying theme is the general (e.g., NumA1, eIF1, Bud31) or localized (e.g., Cbp4a in NoSC1) distribution of proteins linked to cell cycle regulation. Two other proteins linked to cellular stress responses (i.e., Hsp32, FhkA) localize to nucleolar subcompartment NoSC3 which could imply a localization of stress-related functions. Clearly, much remains to be done to prove the significance of the identified nucleolar subcompartments in *Dictyostelium*.

## 4. Changes in Nucleolar Number and Positioning

The nucleolus of *Dictyostelium* undergoes significant changes in shape, location and number in the transition from growth to development [10,11]. While 2-4 nucleoli characterize growth phase cells, this number diminishes to 1–2 during aggregation where one nucleolus resides in a nozzle-like nuclear protrusion that points in the direction of cell migration. With time only this single, microtubule-dependent, nozzle-localized nucleolus remains. These nucleolar events coincide with the turnover of 75% of the growth phase rRNA during development [19].

Nucleoli are excluded from the centrosomal region showing a preferential localization that is opposite to its location [10]. Nucleolar positioning as well as nucleolar size and number are directly linked to the level of ongoing ribosome production [20]. Clearly there are differences between growth phase and developmental nucleoli. A nucleolar mutant in *Dictyostelium* that only forms nucleoli during growth progressively loses its nucleolus during development but is unable to generate the developmental nucleolus [21]. This “anucleolate” mutant is unable to complete development but refeeding of the cells induces the reformation of growth phase nucleoli. The underlying reasons for this developmental deficiency remain to be studied but might indicate that an early evolutionary stage was the physical separation of interphase/mitotic nucleolar function from specific developmental nucleolar roles. Finally, nucleolar breakdown is one of the earliest events detected in the events of cell death associated with stalk cell differentiation in *Dictyostelium* suggesting that this event might be a common initial step in developmental cell death in other organisms [22].

## 5. The Nucleolar Proteins of *Dictyostelium*

In the following sections, each of the characterized nucleolar proteins is discussed in order of its discovery as a resident of the nucleolus. These short descriptions encapsulate the essence of each protein’s role as a nucleolar protein. In each section, binding proteins, the presence of nucleolar localization signals and the translocations of the proteins during mitosis will be summarized as a prelude to summarizing the data and/or drawing broader conclusions about them. During mitosis in mammals, nucleolar proteins typically redistribute to other cellular locales upon dissolution of the nucleolus [2,23]. In *Dictyostelium*, which undergoes a semi-closed mitosis, the pattern of microtubule localization is typically used to determine mitotic stages [9,24,25]. To date, only five studies have been done on nucleolar protein translocations during mitosis in *Dictyostelium*.

### 5.1. Heat Shock Protein 32 (Hsp32)

The first “resident” nucleolar protein to be identified in *Dictyostelium* was heat shock protein 32 (Hsp32) [26]. Colocalizing with rDNA as beads on a string around the periphery of the nucleolus in unstressed cells, during heat shock it redistributes throughout the nucleolus and nucleoplasm [13,26]. Extended periods of heat shock produce a nucleolus with more pronounced rDNA beads revealing that the structure of the nucleolus in *Dictyostelium* responds to stress, as it does in other organisms [26,27]. In keeping with this idea, heat shock treatment of *Dictyostelium* induces a redistribution of Hsp32 from the nucleolar periphery and Snf12 from the nucleoplasm to nucleolus, as discussed below. Attempts to knock out the Hsp32 gene have failed suggesting it may be a critical protein [26].

Due to the presence of a highly acidic region rich in aspartic acid (asp) and glutamic acid (glu) residues, Hsp32 shares sequence similarity to nucleophosmin (NPM1) and nucleolin, both highly conserved nucleolar proteins in mammals [28]. The highly acidic regions are a common feature thought to be responsible for binding to basic ribosomal proteins and to NLSs of other proteins [29,30]. The role of the acid rich glu/asp region is discussed again for NumA1 below. Hsp32 also binds with high affinity to DNA but this association is not involved in its localization [26].

Hsp32 possesses a monopartite and bipartite NLS, thus sharing similarity with nucleolar proteins from other species [26,31,32]. However, an NoLS has not been identified and the means by which Hsp32 localizes to the nucleolus remains to be elucidated.

### 5.2. Eukaryotic Translation Initiation Factor 6 (eIF6)

Eukaryotic translation initiation factor 6, eIF6, was originally identified as a nucleolar protein based on its localization as peripheral patches in DAPI-stained nuclei plus its sensitivity of actinomycin D treatment). [33]. This highly conserved protein is essential to the production of 60S ribosomal subunits serving as a rate-limiting step in the cell cycle [2,34,35,36]. The pathogenesis of two forms of leukemia—inherited Shwachman-Diamond syndrome (SDS) and sporadic SDS—involve a common pathway in 60S-subunit maturation and the functional activation of ribosomes [37,38]. eIF6 is involved in 60S-subunit maturation and thus could play a central role in the disease process. Since eIF6 is shared by eukaryotes and archaea, *Dictyostelium* serves as a model to detail its function. For example, Weiss et al. [39] used single-particle cryo-EM to dissect the mechanism by which eIF6 gets released from nascent 60S ribosomal subunits in *Dictyostelium*. As in other species, eIF6 prevents 60S maturation by blocking the binding of essential maturation factors and, thus, must be removed for functional ribosomal formation to occur. This conserved mechanism involving eIF6 release is impaired in both inherited and sporadic leukemias.

The N-terminal region of eIF6 contains both an NLS and NoLS with a second potential NLS in the C-term [33]. Scott et al. [40] published an “Experimentally determined NoLS” and a predicted NoLS for eIF6, the latter generated from a program they compiled. However, based on their information summarized in Table 2 of their publication, they erroneously analyzed the sequence of NumA1 not eIf6, albeit with additional errors. Using their program to analyze eIF6 in fact detects no NoLS for this nucleolar protein (O’Day, unpublished results). So, although the NoLS in eIF6 has been mapped to a subdomain, the precise location of the NoLS for eIF6 remains to be revealed. Deletion of the eIF6 gene is lethal in *Dictyostelium* as it is in other species [34,35,36]. Examination of the data from Sillo et al. [41], from their study of genes linked to phagocytosis, revealed that eIF6 is upregulated by factors that induce phagocytosis in *Dictyostelium*.

### 5.3. Tumor Necrosis Factor Receptor-Associated Protein 1 (TRAP1)

Tumor necrosis factor receptor-associated protein 1 (TRAP1) is a member of the Hsp90 family [42,43]. It is a multifunctional protein linked to cell cycle progression, cell differentiation, and apoptosis. Found in the outermost layer of the spore coat it is believed to protect these dormant structures from physicochemical stresses [44,45]. While TRAP1 is a questionable nucleolar protein, there is some evidence it localizes there as well as in other cellular locations including mitochondria to where it translocates during early differentiation [45]. It was first observed in intranuclear patches reminiscent of nucleoli but not verified through actinomycin D or other treatments. As expected, TRAP1expression is induced by heat shock and like other members of the Hsp90 family null mutations are lethal [42,43]. More recently, TRAP1 has gained attention as a protein that can help with understanding mitochondrial diseases [46].

### 5.4. Nucleomorphin A1 (NumA1)

The nucleolar localization of a group of well-established calmodulin-binding proteins from other organisms such as calcineurin, CaM kinase II, and myosin light chain kinase supports a role for calmodulin in the nucleolus [47,48]. In keeping with this, this calcium-sensor and -effector has also been shown to bind and localize in mammalian nucleoli [49,50]. Nucleomorphin isoform NumA1 represents the only verified nucleolar calmodulin-binding protein in *Dictyostelium*. Acting as a regulator of nuclear number and interacting with calmodulin in a Ca^2+^-dependent manner, it is predominantly a nucleolar protein with secondary nucleoplasmic localization [51,52,53]. This pattern of localization is a common feature of nucleolar proteins (e.g., nucleophosmin, adenosine deaminases, and murine double minute 2) that shuttle between the nucleolus and nucleoplasm [54,55].

Full length nucleomorphin contains a breast cancer carboxy-terminus domain (BRCT) that is found in cell cycle checkpoint proteins in other organisms [51,56]. The presence of a highly acidic, glu/asp domain is a common feature of nucleolar proteins including nucleophosmin, nucleoplasmin, nucleolin, and Hsp32 [26,28]. Overexpression of GFP-NumA1 lacking its palindromic glu/asp or DEED domain results in multinuclearity fitting with NumA1’s involvement in cell cycle regulation [51]. In contrast to the regulation of nuclear number in human cells where over 100 proteins appear to be involved, this is the only protein so far linked to this function in *Dictyostelium* [20].

A large number of attributes of NumA1 suggest it is a functional equivalent of the mammalian nucleolar protein nucleophosmin (NPM1) [51]. NPM1 has diverse functions including a role in DNA repair, centrosome duplication and cell proliferation. Mutations in NPM1 are a major cause of acute myeloid leukemia (AML) being present in 20–30% of the cases [57]. The acidic glu/asp domain of NPM1 is involved in histone binding but this function has not been studied in NumA1. The DEED domain of NumA1 is sufficient to target FITC to the nucleus thus acting as an unconventional NLS [51]. Furthermore, NPM1 can act as both a proto-oncogene and as a tumor suppressor [58]. The study of NumA1 thus has the possibility of offering additional insight into the mode of action of NPM1. For example, as covered in the next section, its DEED domain binds Cbp4a in a calcium-dependent manner suggesting NumA1 could be involved in recruiting other proteins to the nucleolus, a mechanism apparently not yet studied for NPM1.

Yeast two hybrid studies revealed that NumA1 interacts with the calcium-binding protein Cbp4a and Zn^2+^-metallopeptidase puromycin-sensitive aminopeptidase A (PsaA) which, in other species, is associated with cell cycle progression and several human diseases including Huntington’s and Alzheimer’s disease [59]. *Dictyostelium* PsaA is similar to PSA from *Drosophila*, mouse, and human. In contrast to Cbp4a (see below), PsaA does not localize to the nucleolus but colocalizes with NumA1 in the nucleoplasm independent of Ca^2+^/Calmodulin [60]. The functional relationship between Cbp4a and NumA1 is strengthened by their apparent co-regulation by developmental morphogens where differentiation factor-1 (DIF-1) upregulates them while cyclic AMP and ammonia leads to their downregulation [61].

An attribute of some nucleolar proteins is the presence of multiple NLSs. NumA1 contains four identified NLSs [51,62]. Three of them (NLS-1, -2, and -4) reside within N-terminal residues 1-120: NLS1, ^31^PKSKKKF^37^ and NLS-1 and -4 within ^48^KKSYQDPEIIAHSRPRK^64^ (NLS-4 is underlined). NLS-3 is found in the C-terminus (^246^PTKKRSL^252^). Multiple NLSs may function to modulate the amount of the nuclear-localized protein, as seen for human TBX5, nucleolar ribosomal protein L7a and transcription factor Nrf2 [63,64,65]. FITC-peptide constructs showed each of these sequences, including the terminal RPRK sequence of the bipartite NLS (48–64), localize to nucleoli revealing the peptide sequences are all NoLS/NLS (e.g., Figure 1) [60]. As with the localization of GFP-NumA1 the nucleolar localization was abolished in the presence of actinomycin D but unaffected by treatments with calcium chelators or calmodulin antagonists. Hence these sequences serve as joint NLS/NoLS for the targeting of NumA1. NoLSs, like NLSs, are typically rich in basic residues but, in the absence of a nucleolar envelope, rather than serving as transport signals they appear to function more as retention signals [66]. Human proteins containing NLS/NoLSs include human NF-κB-inducing kinase, the novel human nucleolar protein phosphatidylinositol 4-kinase and others [66,67].

In addition to NumA1, mitotic translocations have only been studied for four other proteins—Cbp4a, Src1, Snf12 and FhkA—as discussed in the following sections (Figure 3). During prophase nucleoli become indistinct with NumA1 appearing in smaller accumulations adjacent to the inner nuclear envelope as well as being associated with the centrosome. By metaphase these inner membrane accumulations disappear and, for the rest of mitosis, NumA1 appears throughout the cytoplasm with a border of protein adjacent the outer nuclear envelope plus continued centrosomal localization.

### 5.5. Calcium Binding Protein 4a (Cbp4a)

Studies using yeast two hybrid and coimmunoprecipitation, identified calcium binding protein 4a, (Cbp4a) as a nucleolar Ca^2+^-dependent, NumA1-binding partner [52]. This binding occurs via the glu/asp or DEED domain. Addition of actinomycin D leads to a loss of nucleolar CBP4a as does calcium-chelation with BAPTA-AM, supporting Cbp4a as a nucleolar protein that requires calcium for its localization. Fitting with its association with the cell cycle protein NumA1, CBP4a has a putative forkhead-associated domain that is present in numerous cell cycle proteins. *Dictyostelium* possesses 13 calcium-binding proteins (CBPs) including CaM, a major Ca^2+^ effector in all eukaryotes [68,69,70]. In contrast to CaM, calfumirin, and CBP3, the function of the other CBPs remains to be researched [53].

Cbp4a residues ^40^KKCK^43^ have been verified as a true NLS but not an NoLS since FITC-bound peptides show nuclear but not nucleolar localization. In total the results indicate that Cbp4a localizes to the nucleolus not via an NoLS but via the calcium-dependent binding to the DEED domain of NumA [60]. This would suggest that Cbp4a diffuses into the nucleolus to be held there through NumA1-binding. The relationship and behavioral differences between these two proteins was revealed during the events of mitosis where they were the first two nucleolar proteins to be studied during mitosis in *Dictyostelium*.

During mitosis CBP4a reveals a unique distribution that suggests the presence of previously undetected intranuclear subdomains that persist throughout the mitotic stages (Figure 3) [71]. During prophase, nucleolar dissolution is accompanied by the accumulation of CBP4a as multiple, discrete nucleoplasmic accumulations called “CBP4a islands”. In addition to these multiple smaller accumulations, during metaphase two larger islands localize to the metaphase plate region. Through anaphase and telophase, these accumulations migrate to the inner membrane as if in anticipation of reforming post-mitotic nucleoli. To date, no other nucleolar or nuclear protein has shown this sequence of events during mitosis. If the nucleolar binding of Cbp4a is dependent on diffusion, then retaining this protein within the nucleus may be essential to ensuring there is enough protein present when the nucleolus reforms during telophase.

### 5.6. SWI/SNF Complex Component SNF12 Homolog (Snf12)

SWI/SNF is a nucleosome remodeling complex, composed of 9–12 proteins called BAFs (Barrier-to-autointegration factor), highly conserve proteins that regulate gene transcription [73]. The complex mediates multiple other processes including cell proliferation, differentiation and DNA repair. It serves as a tumor suppressor by regulating the p53-mediated transcription of cell cycle genes. BAF60a (Snf12 in yeast) mediates its interaction with p53. *Dictyostelium* Snf12 is a predominately nucleoplasmic protein that localizes to nucleoli in ~20% of cells, as seen in mice [74,75]. Appropriately, it possesses conserved SWIB and COG domains found in BAF proteins, but these are not involved in nucleolar positioning. Instead an experimentally defined NLS/NoLS (^372^KRKR^375^) defines both its nuclear and nucleolar localization.

Unexpectedly, treatment of cells with actinomycin D increases the levels of nucleolar Snf12 which leads to an outward bulging of nucleoli followed by the cytoplasmic accumulation of Snf12-rich vesicles. Heat shock treatment also leads to a major increase in the nucleolar localization of Snf12. The rapid increases in nucleolar localization after heat shock and actinomycin D treatment, suggests Snf12 may function in the stress response. As mentioned above, heat shock treatment of *Dictyostelium* induces a redistribution of Hsp32 from the nucleolar periphery [26]. This is not surprising since the nucleolus is the central hub for coordinating the response to cell stress in other species where the composition of the nucleolus is stress-dependent [76]. In keeping with the results summarized here, heat shock and AM-D treatment both cause nucleolar accumulation of specific human proteins [77].

Snf12 undergoes several translocations during mitosis (Figure 3). With nucleolar dissolution during prometaphase, it first shifts from its nucleolar locale to take up a predominantly nucleoplasmic location with some localization in the cytoplasm. During metaphase and through anaphase it then exhibits a relatively uniform cellular distribution before reacquiring its nuclear/nucleolar localization during telophase.

### 5.7. Forkhead-Associated Kinase Protein A (FhkA)

Rad53 (CHK2 in humans) is a tumor suppressor protein involved in DNA damage (genotoxic) stress response [78]. It is recognized as a nuclear protein that possesses a C-terminal bipartite NLS [79]. The *Dictyostelium* Rad53 homologue forkhead-associated kinase protein A, FhkA, is a nucleolar protein. Immunolocalization shows is resides at the periphery of the nucleolar patches (i.e., NoSC3) being more concentrated adjacent to the nuclear envelope. Like Snf12, actinomycin D treatment leads to its nuclear expulsion as nucleolar protein-containing vesicles that end up in the cytoplasm. Its mitotic dynamics are also fitting for a nucleolar protein, yet its specific function there remains to be elucidated. During mitosis FhkA redistributes throughout the cell with an enhanced level of localization evident adjacent to the nuclear envelope from prometaphase through telophase (Figure 3). Like NumA1, FhkA also localizes within the spindle fiber region.

### 5.8. Bud31

As one of the last nucleolar proteins to be identified so far, less is known about Bud31 than any of the others. A comparative study of spliceosomal genes in *Dictyostelium discoideum* identified Bud31 but no further analysis of the protein was carried out [80]. In yeast where the protein was first identified, Bud31 is involved in cell cycle regulation, specifically functioning at the G1/S regulatory or start point [81]. A search of dictyBase.org indicates Bud31 is a putative RNA splicing factor or transcription factor. While the gene has been identified in humans and other species, its function in *Dictyostelium* has not been studied. Selected as a nuclear protein for comparison, it was shown that Bud31 localized throughout the *Dictyostelium* nucleolus along with NumA1 and eIF6 (Figure 2) [18]. However, its location during mitosis and the presence of an NoLS have not been assessed.

### 5.9. Src1

*Dictyostelium* Src1 is homolog of the helix-extension-helix family that localizes adjacent to the inner nuclear membrane [72]. Because of its interaction with the major nuclear lamina protein NE81, its juxtaposition to the inner nuclear membrane and its unchanging location during the cell cycle, Src1 has attributes of an inner nuclear membrane protein that is involved in nuclear lamina formation. In other species, Src1 is also implicated in nucleolar organization through its ability to stabilize repetitive rDNA sequences [82]. GFP-Src1 and immune-transmission electron microscopy reveal that Src1 is an inner nuclear membrane protein that is tightly linked to the positioning of nucleoli in *Dictyostelium* [72]. Future work will have to be done to verify the significance of this relationship and whether Src1 fits the description of a true nucleolar protein. Since Src1 retains its localization throughout mitosis, this puts it in a position to serve as part of the reformation points for nucleolar reassembly during telophase (Figure 3).

## 6. Dozens of Unconfirmed Nucleolar Proteins

Considering that human nucleoli appear to contain over 4500 proteins that are involved in a myriad of essential cell functions with many linked to various diseases, studies on the population of nucleolar proteins of *Dictyostelium discoideum* are still in their infancy. In addition to the proteins discussed above many others, as side issues from studies not directly related to nucleolar structure and function, have been linked to but not yet proven to reside in its nucleolus.

Based on GFP fluorescence images, Meier et al. [83] suggested that RbdB, a nuclear double-stranded RNA binding protein, accumulates in nucleolar foci along with Dicer B, an RNase. However, no experimental validation of this nucleolar localization has been undertaken. What’s more GFP-RbdB foci in that publication appear only at a few nucleolar edges as well as at nuclear periphery away from nucleolar regions, suggesting this association may be random. It will be important to determine if actinomycin D treatments will alter these focal locations or if either RbdB or Dicer B have NoLSs that can be deleted to alter any nucleolar association.

Ase1 is a microtubule cross-linking protein with two homologs (A, B) in *Dictyostelium* [84]. Ase1A shares moderate similarity to the human PRC1 isoform and, similarly, contains two NLSs (320PIEKLKK327, 612PNNKKK1618). During interphase GFP-Ase1A localizes within the dense patches adjacent to the inner nuclear envelope. During mitosis, the protein becomes distributed in the nucleoplasm before localizing within the spindle. The intense and precise localization of Ase1A as a single dot within the interphase nucleolus of *Dictyostelium*, and its translocations during mitosis, make it a strong candidate as a valid nucleolar protein [84]. Ase1 appears to localize in NoSC2, the nucleolar residence of Snf12, but any colocalization remains to be verified.

A search for nucleolar proteins of *Dictyostelium* at the Uniprot website (www.uniprot.org) generated a list of over 60 putative nucleolar proteins that were identified as such based on molecular similarity, function or process. For example, multiple low molecular weight ribonucleoproteins and snRNA associated proteins dominated the list. Also included were HEAT-repeat containing proteins, processome components, GTP-binding proteins and others including unknown proteins. Nop56 is another putative nucleolar protein for which antibodies are available online but its nucleolar localization has not been validated. While these proposed constituents support the universality of identified nucleolar functions in *Dictyostelium*, without further analyses, how they affect its overall structure and function and how they might translocate during mitosis or stress remain to be investigated.

## 7. Protein Associates of Nucleolar Proteins

Studying proteins that interact with nucleolar proteins can provide additional insight into their functions in situ or when they translocate to different cellular locales. For *Dictyostelium*, this area of research has only just begun. Yeast two hybrid and co-immunoprecipitation analyses revealed that NumA1 not only binds to nucleolar Cbp4a but also to nuclear PsaA. PsaA shares the critical domains of human Psa including GAMEN and Zinc-binding domains and, similarly, is also inhibited by bestatin methyl ester (BME) [60,85]. DdPsaA has been proven to possess defined NLS and NES sequences. Those studies also revealed the importance of on DdPsaA in both cell proliferation and cell differentiation coinciding with the central functions proposed for NumA1. These tasks were further supported by the finding that DdPsaA in turn binds to cyclin-dependent kinase 5 (DdCdk5) [86]. In support, the Cdk inhibitor roscovitine was effective on DdCdk5 activity and dose-dependently inhibited cell proliferation [86]. Since the understanding of the relationship between nucleolar integrity and cell cycle progression remains to be elucidated in any organism, defining the interplay between NumA1, DdPsaA and DdCdk5 during cell proliferation should be especially enlightening.

## 8. Nucleolar Prion-like Proteins

Proteins possessing prion-like domains (PLDs) are implicated in numerous protein-misfolding diseases, especially neurodegenerative diseases [87]. Enriched in glycine and polar amino acids, PLDs are low complexity amino acid sequences often found in RNA-binding proteins. In the nucleus, these PLD proteins aggregate with other proteins as discrete paraspeckles. *Dictyostelium* not only has a Q/N-enriched proteome, it also has the highest amount of prion-like proteins of the organisms studied so far [88]. Overexpression of human huntingtin exon 1 or yeast prion protein Supp35 does not lead to the expected formation of toxic cytosolic aggregates but instead produces harmless, soluble proteins [88]. However, disruption of molecular chaperone function causes these proteins to form insoluble cytotoxic assemblages. Of relevance here is that small accumulations of the huntingtin and yeast prion-like proteins localize to nucleoli suggesting its role in regulating these events in *Dictyostelium*. Similar huntingtin aggregates have been discovered adjacent to human nucleoli [89]. Other studies on the huntingtin in *Dictyostelium* that reveal that the mutant protein does not form aggregates while the normal protein is involved in multiple cellular processes including growth, cation homeostasis, cell motility, cell shape, chemotaxis, cell-cell adhesion, cell fate determination and osmoregulation [6]. Understanding how *Dictyostelium* prevents prion-like protein aggregation, and the role of the nucleolus in this function, could lead to therapies for preventing the formation of toxic plaques found in Huntington, Alzheimer’s, Parkinson’s and other prion-based neurodegenerative diseases [90].

## 9. Conclusions and Questions

Once considered to be homogeneous structures, the nucleoli of *Dictyostelium discoideum* display specific protein localizations suggestive of nucleolar compartmentalization. Nucleolar targeting signals have been identified that target proteins to either the whole nucleolus or specific subcompartments within it and, thus, may have biomedical uses. Other studies of nucleolar proteins have aided in the understanding of mitosis in the social amoebozoans revealing that it is a semi-closed rather than closed mitotic event. Despite these relatively unique aspects that offer valuable evolutionary insights into the nucleolus, the presence of nucleolar proteins that share structural and functional similarities to their mammalian counterparts suggests that *Dictyostelium* can be a useful system for research into nucleolar-related diseases thus adding to its value as a model organism for biomedical research.

Following the translocation of nucleolar proteins during mitosis, when nuclear dissolution/reformation occurs, has led to some interesting discoveries not only about the paths of the proteins but about the nature of *Dictyostelium* mitosis. For the latter, the apparently intact nuclear envelope becomes permeable revealing that a semi-closed, not closed, type of mitosis is occurring [25]. As might be expected, different nucleolar proteins demonstrate different patterns of dispersal and intracellular localization during mitosis. Whether this has anything to do with a mitotic function that complements their nucleolar roles, remains to be determined. The unique events that occur after treatment with actinomycin D are also interesting. Why do some nucleolar proteins simply translocate individually from nucleoli to other cellular locales after actinomycin D treatment while others remain associated in what could be termed a “nucleolar protein export vesicle”?

The nucleolus of *Dictyostelium discoideum* is linked to various diseases and abnormal cellular states. It is involved in the response to cellular stress and has gained attention in the study of various neurodegenerative and other diseases some of which were addressed above [6,90]. In humans, abnormal nucleolar size, structure and function are associated with neurodegeneration, various cancers and other diseases [20]. As with any research, many questions remain. *Dictyostelium discoideum* possesses multiple nucleoli that vary in size and number (2–4) during interphase and then decrease in number to a single nucleolus during development. Is there a dominant nucleolus that determines if more are needed during growth then directs their disappearance during development? What are the differences between the multiple vegetative nucleoli and the single developmental nucleolus?

The discovery of Src1 adds to an already interesting developing story. Is this inner nuclear matrix-binding protein the attachment and organization site for nucleolar localization. Its tight association between nucleoli and the inner nuclear membrane plus its persistent localization through the cell cycle implicates it in this function. Whether or not it serves this function, what are the first nucleolar proteins to start nucleolar reformation during telophase? Would it be one from the nucleoplasmic retained nucleolar proteins or one from those that translocated to the cytoplasm during the semi-closed mitosis. Because of their size, the Cbp4 islands that persist in the nucleoplasm could be sites of storage for some nucleolar proteins during mitosis, ready for nuclear reformation during telophase. While there is a clear relationship between nucleolar proteins and the centrosome, what role does the later play in the formation of the former? The answers to these and previous questions could be useful in advancing *Dictyostelium’s* role as a model research organism as well as providing insight into the evolution of eukaryotic nucleoli.

Several nuclear and nucleolar proteins have recently been suggested to be promising targets for anti-cancer drugs [90]. Despite this, the targeting of proteins to the nucleolus is still not fully understood. As focus on the nucleolus increases with the understanding of its importance in a diversity of diseases, understanding the targeting of nucleolar proteins is of key importance since it will serve as a guide to developing vehicles for pharmaceutical delivery. The NLS/NoLS peptides of NumA1 could be of biomedical value since the conjugation of peptides to drugs to target their delivery to specific subcellular locales often results in enhanced efficacy coupled with decreased side effects.

Nuclear localization signals (NLSs), that also act as nucleolar localization signals (NoLSs), have been identified for two nucleolar proteins (i.e., NumA1 and Snf12) in *Dictyostelium*. These NLS/NoLSs represent the first NoLSs and first NLS/NoLSs identified in this amoebozoan. A perplexing aspect of nucleolar protein localization is how some NLSs, that share sequence similarity to other strict NLS, can also serve as functional NoLSs. It has not been possible in any species to precisely define what the critical and essential attributes of an NoLS are. That said, all four of the NLSs in NumA1 (PKSKKKF, KKSYQDPEIIAHSRPRK, RPRK and PTKKRSL) can localize FITC to nucleoli revealing these sequences are all NoLS/NLS [51,62]. Thus, they potentially could serve as vehicles for general delivery to the nucleolus. In contrast, FITC-KRKR localizes to NoSC2, demonstrating that this NoLS not only localizes to the nucleolus but to a specific region within it. The identification of NLS/NoLS-binding proteins could provide some insight into whether these peptide sequences would be useful for drug delivery in other organisms.

## Figures and Tables

**Figure 1 cells-08-00167-f001:**
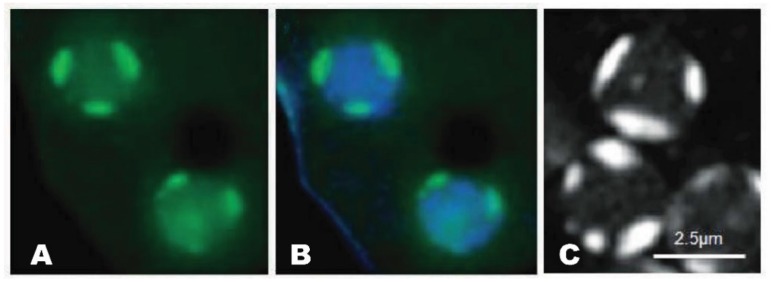
Nucleolar patches of *Dictyostelium*. **A**. GFP-NumA1 (green) localization. **B**. GFP-NumA1 localization with DNA (Hoescht, blue staining) stained. **C**. FITC-NumA1-NLS1 localization (white patches).

**Figure 2 cells-08-00167-f002:**
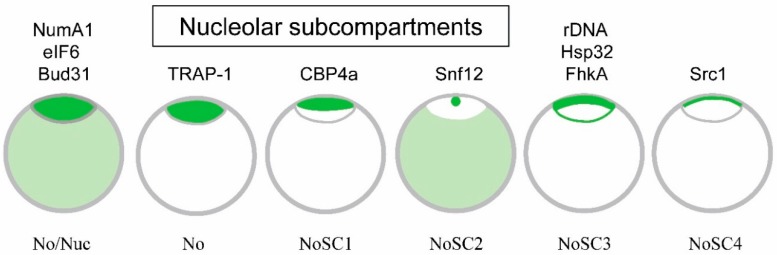
Localization of *Dictyostelium* nucleolar proteins. Different *Dictyostelium* nucleolar proteins localize differently as indicated by the green staining. The intensity of the staining summarizes their general differential localization in the nucleolus versus the nucleoplasm. NumA1, eIF6, and Bud31 localize to both the nucleolus and nucleoplasm (No/Nuc), TRAP-1 localizes only to the nucleolus (No), CBP4a localizes only to NoSC1, Snf12 localizes to NoSC2 as well as the nucleoplasm, while Hsp32 and FhkA localize to the nucleolar periphery, possibly representing NoSC3. Src1, a homolog of the helix-extension-helix family, is a questionable nucleolar protein that localizes to a region tentatively labelled NoSC4. The grey lines serve only to indicate the borders of the nucleolus and the nuclear envelope. (modified and updated after [18]).

**Figure 3 cells-08-00167-f003:**
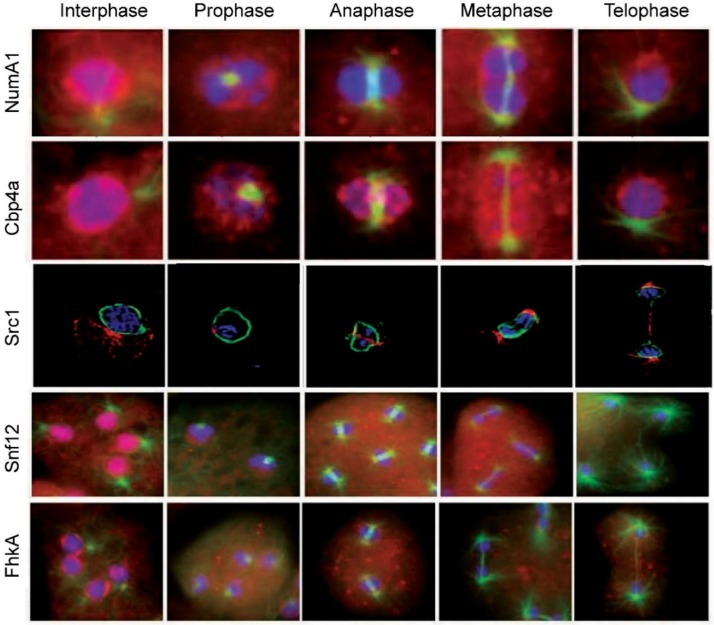
Nucleolar protein translocations during mitosis in *Dictyostelium discoideum*. NumA1, nucleomorphin A1, a cell cycle protein; Cbp4a, calcium-binding protein 4a, a NumaA1 binding protein; Src1, helix-extension-helix family homolog; Snf12, a nucleosome remodeling complex component; FhkA, a Rad53 (Chk2 in humans) tumor suppressor homolog. Note: the images for Src1 are from [72] with modifications

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
