# Peer review of "Proteins of the Nucleolus of Dictyostelium discoideum: Nucleolar Compartmentalization, Targeting Sequences, Protein Translocations and Binding Partners"

_cells, 2019, doi:10.3390/cells8020167_

Round 1

Reviewer 1 Report

The paper by Danton O'Day is a valuable up-dated review on the nucleolus in Dictyostelium, and as such is also a good source of references on this topic. On a less positive note, this paper also tends to be, perhaps unavoidably, a catalog of the known nucleolus-related molecules. An effort at some synthesis is made in the Discussion.

Minor points: there is a slight tendency to sometimes generalize. For instance, line 23, "to complete the picture", the picture cannot yet be completed since there are many still unknown nucleolar proteins; "to enrich the picture" might be more appropriate. Similarly, line 147, "a universal initial step" might be replaced by "a more general initial step". Also, line 401, "the universality of nucleolar functions in Dictyostelium" is not clear; does this mean that these functions can be found elsewhere than in Dictyostelium, or that the constituents account for all known nucleolar functions in Dictyostelium ?

In spite of these minor points, that the author may want to correct, again the paper is a valuable contribution to this field, and deserves to be published in Cells.

Author Response

We thank the reviewer for their frank assessment of this review.

l. 23, As suggested, “to complete the picture” was changed to “to enrich the picture”

l. 147. “a universal initial step” was changed to “a common initial step”

l. 401, "the universality of nucleolar functions in Dictyostelium" was changed to, "the universality of the identified nucleolar functions in Dictyostelium"

Reviewer 2 Report

This review article is a comprehension of the Dictyostelium discoideum nucleolus proteins, nucleolar compartmentalization, targeting sequences, protein translocations, and binding partners. The review is crafted around the structural organization of nucleolus and its functional role in the context of localized proteins and interaction with other cellular proteins.

Overall the review is well-written except some minor issues can be corrected and published.

Some minor issues and typos for example: 

In Figure 1: Source of the figure (If it is taken from another source then a note can be added) or generated in the lab for the review?

Line 63-64, a full stop placed twice. 

Line 87-88 is confusing to read. Defining abbreviation NoSC1 before the sentences could be better.

Author Response

We thank the reviewer for their frank and positive assessment of this review.

Figure 1: is original so no citation is required

Line 63-64, extra stop removed 

Line 87-88, was rewritten as, “CBP4a localizes to a patch close to the nuclear envelope designated as nucleolar subcompartment 1 (NoSC1).”